# Hormonal Dysfunction in Paediatric Patients Admitted to Rehabilitation for Severe Traumatic Brain Injury: Analysis of the Associations with Rehabilitation Outcomes

**DOI:** 10.3390/children11030304

**Published:** 2024-03-05

**Authors:** Sara Galbiati, Federica Locatelli, Francesca Formica, Marco Pozzi, Sandra Strazzer

**Affiliations:** Scientific Institute IRCCS Eugenio Medea, 23842 Bosisio Parini, Italy; sara.galbiati@lanostrafamiglia.it (S.G.); federica.locatelli@lanostrafamiglia.it (F.L.); francesca.formica@lanostrafamiglia.it (F.F.); sandra.strazzer@lanostrafamiglia.it (S.S.)

**Keywords:** IGF-1, brain injury, biomarker

## Abstract

Traumatic brain injury is often accompanied by defects in hormone levels, caused by either peripheral gland dysfunctions or by an insufficient central stimulation of hormone production. The epidemiology of endocrinological defects after traumatic brain injury is quite well described, but the consequences of hormone defects are largely unknown, especially in paediatric patients undergoing neurological rehabilitation. Only one previous study reported on a cohort of 20 children with traumatic brain injury and found a low incidence of hormone defects and a correlation between some hormone levels and neurological recovery. In this study, we performed a retrospective chart review on patients affected by severe subacute traumatic brain injury. Their levels of cortisol, ACTH, IGF-1, TSH, free T4, free T3, and prolactin were collected and compared with reference ranges; we then used regression models to highlight any correlation among them and with clinical variables; last, we probed with regression models whether hormone levels could have any correlation with clinical and rehabilitation outcomes. We found eligible data from the records of 52 paediatric patients with markedly severe traumatic brain injury, as shown by an average GCS of 4.7; their age was 10.3 years, on average. The key results of our study are that 32% patients had low IGF-1 levels and in multiple regression models, IGF-1 levels were correlated with neurological recovery, indicating a possible role as a biomarker. Moreover, 69% of patients had high prolactin levels, possibly due to physical pain and high stress levels. This study is limited by the variable timing of the IGF-1 sampling, between 1 and 2 months after injury. Further studies are required to confirm our exploratory findings.

## 1. Introduction

Traumatic brain injury (TBI) is a leading cause of morbidity and mortality in children and adolescents. Survivors of severe TBI are more prone to health problems and functional deficits, resulting in poorer school performances and an increased risk of behavioural problems. After a brain injury, 70% of children receiving motor rehabilitation improve their motor functions within 1 year from injury [1]. However, TBI is clinically heterogeneous, and the entity of brain damage may not truly reflect the prognosis of patients with TBI [2]. For this reason, professionals find it difficult to understand how, in the face of similar starting conditions, patients can have different responses to the same rehabilitation project with equivalent intensities and elements of care [3]. In previous work, we were able to demonstrate that there are four data-driven trajectory subtypes based on distinct temporal progression patterns and predict the treatment response at the individual-subject level based on clinical and functional conditions at first discharge [4]. It remains to be discovered what the factors and medical conditions that influence the entity of brain damage are, which also influence the type of response to rehabilitative treatment. Indeed, it would be important to be able to improve the prognosis on arrival from intensive care in order to better support the family in the intensive rehabilitation process.

Among the medical problems that can arise after a TBI, there are alterations of the hypothalamic–pituitary axis. The cause of pituitary dysfunction after TBI is still not well understood, but evidence suggests a few possible primary and secondary causes [5]. Direct damage may be due to a basal skull fracture and subsequent fracture of the Sella, which can directly damage the pituitary gland or may be caused by haemorrhage or an acute infarct in the gland. Secondary damages may be due to hypoxia, brain swelling, hypotension, anaemia, or the stress of critical illness and the possible effects of medications [6,7].

Current evidence suggests that the results of studies focusing on the incidence of pituitary dysfunction after TBI in adults are varied (23 to 69% of the cases) [8], and possible reasons for this could be the difference in patient selection, the severity of injury, the methods used, the study design, and the timing of the assessment, both in acute and recovery phases [5].

Much less is known about neuroendocrine dysfunctions in the paediatric population compared to those in the adult population. However, the scientific literature documented a 16–21% incidence of hypopituitarism at 1–5 years after injury [8,9,10,11,12,13,14], with a single exception, with an incidence equal to 61%, in the work of Niederland and colleagues [15]. These results indicate a lower frequency of hormonal deficiency in children than in adults. Hormonal dysfunction was more likely after a severe injury than after a medium injury [8]. However, although the incidence is low in mild TBI and if a systematic pituitary assessment is not required for them, physicians should monitor children 1 year after mild TBI with particular attention to growth and weight gain [16]. Patients with TBI without neuroendocrine changes and those with TBI-induced hypopituitarism share the same clinical manifestations, such as attention deficits, impulsion impairment, depression, sleep abnormalities, and cognitive disorders [7]. This is because hormones have a direct effect on brain growth and function [17,18] and can affect the quality of life of patients [19]. For this reason, TBI-induced hypopituitarism may be neglected in TBI victims, and it would be expected that underlying hypopituitarism would aggravate the clinical picture of TBI itself. Therefore, the diagnosis and treatment of unrecognised hypopituitarism due to TBI are very important not only in decreasing morbidity and mortality due to hypopituitarism but also in alleviating the chronic sequelae caused by TBI [7].

To our knowledge, only two studies have investigated the relationship between hormonal alterations and outcomes, the first in the acute phase in a mixed paediatric and adult population [20] and the second in the subacute-chronic phase in a paediatric population [13]. Rao and colleagues found that high levels of serum IGF-1, testosterone, and fT4 were associated with improved functional outcomes in children with TBI [13].

The aims of this study, conducted on paediatric inpatients recovering from severe TBI, were to assess the prevalence of various neuroendocrine dysfunctions when they enter rehabilitation and the possible associations with clinical variables and with differences in rehabilitation outcomes.

## 2. Materials and Methods

This was a retrospective observational study, conducted by collecting data from our institutional electronic clinical database and from paper clinical records. All procedures described in this work were carried out following the clinical routine, during the years 2018–2022, and in accordance with the Declaration of Helsinki. The inclusion criteria for the data collection of this study were: being admitted to our rehabilitation unit after severe traumatic brain injury, i.e., Glasgow Coma Scale score (GCS) of 8 or less; age 1–18 years at admittance to rehabilitation; having no known genetic disease or conditions that may predispose to the alteration of hormonal levels; and having recorded results of endocrinological exams performed during the first month in rehabilitation after discharge from intensive care. From the clinical records, we collected the following characteristics of patients: Sex, age, age at TBI, GCS, Glasgow Outcome Score (GOS) at admittance, and discharge from the first rehabilitation stay. We collected the results of endocrinological exams comprising cortisol, ACTH, IGF-1, TSH, free T4, free T3, and prolactin. These exams were performed during the first month of the rehabilitation stay, sub-acutely after TBI. We also collected, from records in the best temporal proximity to endocrinological exams, patients’ weight and BMI, whether they used corticosteroid drugs in the short or long term, whether they had an electroencephalography (EEG) indicating the presence of sleep or partial sleep, and whether they had bone density values indicating osteoporosis. We calculated the GOS improvement after rehabilitation as the difference between GOS scores at admittance and discharge from rehabilitation (dGOS), which indicates recovery when the dGOS value is above zero. We elaborated on the raw values of hormone levels considering normal values by age and sex and analysed the distribution of values within or outside reference ranges, using the mean and standard deviation and analysis of variance. We then analysed the inter-correlation of hormone values, based on a priori physiological hypotheses connecting one hormone with the others, using Pearson correlations. We refined correlations by applying linear regression models that included multiple independent variables. We finally analysed possible correlations between hormone levels and outcomes of interest—in particular, BMI, the presence of sleep, osteoporosis, and GOS improvement after rehabilitation—while correcting for clinical variables including sex, age, and GCS using linear regression models. For all tests, the significance levels were set at *p*-values < 0.05, two-tailed. Analyses were conducted using GraphPad Prism v.9.4.1.

## 3. Results

A total of 52 children and adolescents who suffered from severe TBI were included in this study. The mean patient age was 10.3 ± 5.5 years; 17 were females and 35 were males; their GCS at the event was, on average, 4.7 ± 1.5. The analysis of hormone levels involved with the HPA axis showed average cortisol levels of 471 ± 510 nmol/L and ACTH levels of 26.6 ± 13.4 ng/L. The IGF-1 level was, on average, 231 ± 140 mg/L. Thyroid hormones were almost always in range, with the TSH being, on average, 2.45 ± 1.78 UI/L, fT4 1.3 ± 0.3 ng/dL, fT3 4.15 ± 1.06 pg/mL. Prolactin was almost always markedly elevated (on average, 25.5 ± 13.3 ng/mL); Table 1 reports, for each investigated hormone, how many patients had values below, within, or above the reference ranges. Not all patients were examined for all hormone levels (see the Table 1 caption).

When analysing inter-correlations between hormone levels, apart from the physiologically expected correlations (ACTH with cortisol and ACTH/cortisol; TSH with T4 and T3), we found a significant correlation between levels of ACTH and TSH, with r = 0.30 and *p* = 0.034 (n = 49). We next computed multivariable models: the results are reported in Table 2 and no meaningful change occurred with respect to correlations.

Regarding our clinical outcomes of interest, at admittance to the first rehabilitation stay, patients’ weight was, on average, 33.1 ± 17.0 Kg, and their BMI was, on average, 16.9 ± 3.2. At a polysomnography exam, 14 patients had a regular sleep structure, 10 had a partial sleep structure, and 38 had no evidence of sleep; osteoporosis was evident in 3 patients; at the end of the first rehabilitation stay, 39 patients (75%) had an improvement in GOS, and overall, the GOS change was +1.5 ± 1.4, indicating marked recovery.

Since, in the literature, several endocrine factors are hypothesised to have an effect on recovery from neuronal damage, we performed one last analysis, correlating neurological recovery (dGOS) with hormonal levels, while correcting analyses for clinically relevant factors, including age, sex, and GCS.

The results, with R^2^ = 0.32, indicated that the GCS (β = 0.48, *p* < 0.001) and IGF-1 levels (β = 0.003, *p* = 0.022) were significantly correlated with dGOS change, i.e., neurological recovery. It is interesting to note that the difference in IGF-1 levels between patients in the “below reference” group and those in the “above reference” group was 129 mg/L, together with the β parameter of the above regression (+0.003 dGOS points for each 1 mg/L of IGF-1): patients with IGF-1 levels above the reference would have a better improvement of 0.39 GOS points as compared to patients with IGF-1 levels below the reference. The clinical outcomes of weight, BMI, presence of sleep, and presence of osteoporosis were not correlated with any clinical variable or hormone level investigated.

## 4. Discussion

Neuroendocrine dysfunction after TBI is characterised by a deficiency in at least one endocrine axis, which may be caused by either an inability of the downstream glands to produce hormones or by an insufficient supply of centrally released stimulating hormones. There is a considerable amount of literature focused on the description of neuroendocrine defects after TBI in children; however, to our knowledge, only one previous study focused on the influence that neuroendocrine defects may have on the rehabilitation course of children [13]. Rao and colleagues found that 10% of their paediatric patients had low IGF-1 levels; in our cohort, we documented low IGF-1 levels in 32% of patients. One possible explanation for this discrepancy is that our patients were more severe, all being diagnosed with severe TBI, while the work of Rao and colleagues included only 55% severe patients. Moreover, they collected data from 20 patients, whereas our wider cohort included 52 patients. Another consideration is that serum IGF-1 levels are also dependent on body mass composition and level of fitness, and the BMI of our patients was, on average, low, although we found no meaningful correlation between IGF-1 levels and BMI in the present work.

Regarding the rehabilitation prognosis, Rao and colleagues reported that high levels of serum IGF-1, testosterone, and fT4 were associated with improved functional outcomes in children with TBI; of note, the statistical analyses conducted were single-variable models, not corrected for confounding or inter-correlated factors.

In fact, we could correct our analyses for key confounders, and we also found that IGF-1 levels were positively correlated with rehabilitation recovery, measured by the GOS increase, supporting a possible role as a biomarker for IGF-1.

That high IGF-1 levels may correlate with intense proliferation and remodelling in the brain is not entirely novel: the expression of IGF-1 in the brain is high during development, where it promotes brain growth by supporting neuronal survival, neurite outgrowth, oligodendrocyte maturation, and myelination [21]. Despite the multiple roles IGF-1 plays in brain neuroplasticity, very little is known about the regulation of endogenous IGF-1 after TBI. New studies would be needed to evaluate these aspects in children by correlating hormone levels with clinical aspects and, especially, neurological recovery. Moreover, IGF-1 is under investigation as a potential treatment for improving neurological recovery after brain injury [22,23,24]; however, randomised controlled trials are required in order to clarify its potential usefulness and safety for clinical use. An overview of IGF-1 roles in brain injury is available from Mangiola et al. (2015) [25].

Another significant result of our study is that we found increased prolactin levels in 69% of patients. We found no correlation with cortisol levels or with the presence of sleep patterns, indicating no influence of HPA factors or sleep on prolactin production; however, we believe that the elevated levels of prolactin in our cohort may be suggestive of high stress levels, possibly consequent of physical distress and/or pain [26] that we could not measure reliably in unconscious patients. An alternative explanation may involve a dysfunction of dopaminergic projections towards the pituitary, which would leave prolactin production uninhibited.

The main limitation of our study is the cross-sectional nature of hormone level measurements, which may have led to some misinterpretations of transient abnormalities; in this study, blood sampling for hormone levels was conducted 1 to 2 months after brain injury. For IGF-1 in particular, this can be important, as there can be early rises followed by reductions. It would be valuable, in terms of future comparisons/replication, to provide some greater specificity in terms of the time that blood was drawn relative to the injury. This study had a relatively small sample size, which limits the reliability of the results and the sensitivity of our analyses. Additionally, it had a retrospective, non-randomised design. Moreover, our results must be interpreted in light of the high severity of the cohort of patients we described. One specific limitation regarding endocrinological function is that we did not perform any secretion stimulation test, since they were outside the clinical practice.

## 5. Conclusions

We observed, in a cohort of young patients recovering from severe traumatic brain injury, prolactin elevation and IGF-1 reduction. We found an association between higher IGF-1 levels and better neurological recovery. Prolactin elevation should be further investigated as a potential biomarker of distress, particularly relevant in unconscious patients who cannot communicate. IGF-1 may be a marker associated with better neuronal recovery. Our findings must be verified in larger prospective studies.

## Figures and Tables

**Table 1 children-11-00304-t001:** Hormone levels with respect to reference levels.

Hormone	Below Reference		In Reference Range		Above Reference	
	Patients Number (%)	Hormone Level Mean ± SD	Patients Number (%)	Hormone Level Mean ± SD	Patients Number (%)	Hormone Level Mean ± SD
Cortisol nmol/L	19 (37.2%)	12.5 ± 6.1	21 (41.2%)	16.0 ± 8.1	11 (21.6%)	1119 ± 806
ACTH ng/L	4 (8.2%)	8.7 ± 0.9	43 (87.7%)	23.8 ± 10.1	2 (4.1%)	64.4 ± 2.2
Cortisol /ACTH mol/g	25 (51.0%)	9.9 ± 3.1	22 (44.9%)	25.4 ± 9.6	2 (4.1%)	207.7 ± 200.6
IGF-1 mg/L	16 (32%)	169.8 ± 127.4	27 (54%)	215.0 ± 127.7	7 (14%)	298.5 ± 165.9
TSH UI/L	0	/	49 (94.2%)	2.08 ± 1.14	3 (5.8%)	7.81 ± 1.93
fT4 ng/dL	2 (3.9%)	0.66 ± 0.04	49 (94.2%)	1.30 ± 0.21	1 (1.9%)	2.62
fT3 ng/L	0	/	45 (91.8%)	3.93 ± 0.74	4 (8.2%)	6.62 ± 1.00
PRL mg/L	0	/	16 (30.8%)	12.7 ± 4.5	36 (69.2%)	31.5 ± 11.7

Numbers and percentages refer to the number of patients in each category. Average values and standard deviations refer to all patients within one category. Missing values: cortisol 1; ACTH 3; cortisol/ACTH 3; IGF-1 2; TSH and fT4 none; fT3 3; PRL 2.

**Table 2 children-11-00304-t002:** Variables influencing hormone levels.

Hormone Levels (Dependent Variables)	R^2^ of the Regression Models	Independent Variables Significantly Associated with Hormone Levels	
Cortisol	0.96	ACTH β = 9.24 *p* < 0.001	
ACTH	0.90	TSH β = 4.76 *p* = 0.027	
Cortisol/ACTH	0.98	ACTH β = −0.45 *p* = 0.016	Cortisol β = 0.05 *p* < 0.001
IGF-1	n/a	none	
TSH	0.73	ACTH β = 0.11 *p* = 0.027	
fT4	n/a	none	
fT3	n/a	none	
PRL	n/a	none	

Results from linear regression models are reported, including R-square values and parameter estimates with *p* values. Independent variables inserted in regression models were: sex, age, weight, use of corticosteroid drugs, and, as relevant, cortisol levels, ACTH levels, IGF-I levels, TSH levels, fT4 levels, fT3 levels, and prolactin levels.

## Data Availability

The data presented in this study are available for non-commercial purposes on request from the corresponding author. The data are not publicly available due to privacy issues.

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
