# Peer review of "Hormonal Dysfunction in Paediatric Patients Admitted to Rehabilitation for Severe Traumatic Brain Injury: Analysis of the Associations with Rehabilitation Outcomes"

_children, 2024, doi:10.3390/children11030304_

Round 1

Reviewer 1 Report

Comments and Suggestions for Authors

This retrospective chart review of paediatric patients who had suffered severe traumatic brain injury (TBI) reported on endocrine test findings from tests performed during the first month of rehabilitation.

The main finding related to levels of Insulin-like growth factor -1 (IGF-1) which was correlated with measures of neurologic recovery.

The timing of measurement of endocrine tests, and IGF-1 in particular can be important, as there can be early rises followed by reductions. It would be valuable, in terms of future comparisons/ replication, to provide some greater specificity in terms of the time blood was drawn relative to the injury. I presume that could be relatively easily accomplished.

For table 2, I am unclear what model the ‘Model R2 ‘ relates to. If the r for the correlation between ACTH and TSH is 0.30, it seems odd that the Model R2  is 0.90. Should this be 0.09? Can the investigators make more clear what the table is representing? What is the dependent variable. What variables are entered into the regression?  

What is 'R2' (line 153).

Line 196 end a two-line paragraph. The following paragraph seems to address the significance of the prolactin elevation (mentioned in the above-mentioned brief paragraph) so I don’t think it should be a new paragraph (line 197 would be clearer if it was stated that there was no significant correlation of prolactin with cortisol levels…).

The investigators may wish to consider citing ‘Role and importance of IGF-1 in traumatic brain injuries’ Mangiola et al  Biomed research International 2015 (I am not an author) as it provides more context for the findings presented.  

Comments on the Quality of English Language

Overall quite reasonable.

Author Response

Reviewer 1

This retrospective chart review of paediatric patients who had suffered severe traumatic brain injury (TBI) reported on endocrine test findings from tests performed during the first month of rehabilitation.

The main finding related to levels of Insulin-like growth factor -1 (IGF-1) which was correlated with measures of neurologic recovery.

The timing of measurement of endocrine tests, and IGF-1 in particular can be important, as there can be early rises followed by reductions. It would be valuable, in terms of future comparisons/ replication, to provide some greater specificity in terms of the time blood was drawn relative to the injury. I presume that could be relatively easily accomplished.

We thank the Reviewer for the valuable suggestion: we inserted this issue as a limitation of our study (line 207).

For table 2, I am unclear what model the ‘Model R2 ‘ relates to. If the r for the correlation between ACTH and TSH is 0.30, it seems odd that the Model R2  is 0.90. Should this be 0.09? Can the investigators make more clear what the table is representing? What is the dependent variable. What variables are entered into the regression? 

R2 is the R-squared value of the regression model: it describes how much of the overall variability of the dependent variable is explained by the independent variables inserted into the model. For the regression model between ACTH and TSH R2=0.90 means that 90% of the variability in ACTH levels are explained by the regression model, which does not contain only TSH. We have now reported the full list of independent variables and clarified our methods at lines 109 and 140 and in the column headings of Table 2.

What is 'R2' (line 153).

This was a typing error: we corrected R2 to R2. See the response above for the meaning of R2.

Line 196 end a two-line paragraph. The following paragraph seems to address the significance of the prolactin elevation (mentioned in the above-mentioned brief paragraph) so I don’t think it should be a new paragraph (line 197 would be clearer if it was stated that there was no significant correlation of prolactin with cortisol levels…).

We agree with Reviewer and removed the paragraph separation.

The investigators may wish to consider citing ‘Role and importance of IGF-1 in traumatic brain injuries’ Mangiola et al  Biomed research International 2015 (I am not an author) as it provides more context for the findings presented. 

We thank the Reviewer for the valuable suggestion: we inserted this in the discussion (line 195).

Reviewer 2 Report

Comments and Suggestions for Authors

This paper raises a question about the endocrinological defects following traumatic brain injury (TBI) in pediatric patients. In this study, patients aged 1-18 years had severe subacute traumatic brain injury. The authors measured levels of cortisol, ACTH, IGF-1, TSH, free T4, free T3 and prolactin. Through regression models, they seek to identify any significant correlation between these hormones and various clinical variables, also between hormones and clinical and neurological outcomes. The findings suggest a large part of the pediatric patients exhibited abnormal levels of IGF-1 (possible biomarker) and prolactin, which may be linked to their neurological recovery and the stress and pain associated with their condition, respectively.

According to Table 1, hormone levels are measured in different numbers of patients (Cortisol is measured in 52 patients, ACTH in 49, and IGF-1 in 50). Please explain this difference in number of patients. Please add the explanation in the text of the paper.

Author Response

Reviewer 2

This paper raises a question about the endocrinological defects following traumatic brain injury (TBI) in pediatric patients. In this study, patients aged 1-18 years had severe subacute traumatic brain injury. The authors measured levels of cortisol, ACTH, IGF-1, TSH, free T4, free T3 and prolactin. Through regression models, they seek to identify any significant correlation between these hormones and various clinical variables, also between hormones and clinical and neurological outcomes. The findings suggest a large part of the pediatric patients exhibited abnormal levels of IGF-1 (possible biomarker) and prolactin, which may be linked to their neurological recovery and the stress and pain associated with their condition, respectively.

According to Table 1, hormone levels are measured in different numbers of patients (Cortisol is measured in 52 patients, ACTH in 49, and IGF-1 in 50). Please explain this difference in number of patients. Please add the explanation in the text of the paper.

In the caption of Table 1 we had already reported the numbers of missing values for each parameter. We have now specified this also in the text (line 125).

Round 2

Reviewer 1 Report

Comments and Suggestions for Authors

My concerns have been adequately addressed.

Comments on the Quality of English Language

OK

Author Response

Thank you